# Bta-miR-106b Regulates Bovine Mammary Epithelial Cell Proliferation, Cell Cycle, and Milk Protein Synthesis by Targeting the *CDKN1A* Gene

**DOI:** 10.3390/genes13122308

**Published:** 2022-12-07

**Authors:** Xin Wu, Jinfeng Huang, Yanan Liu, Houcheng Li, Bo Han, Dongxiao Sun

**Affiliations:** Key Laboratory of Animal Genetics, Breeding and Reproduction of Ministry of Agriculture and Rural Affairs, National Engineering Laboratory of Animal Breeding, Department of Animal Genetics and Breeding, College of Animal Science and Technology, China Agricultural University, Beijing 100193, China

**Keywords:** bta-miR-106b, *CDKN1A*, BMECs, cell proliferation, milk protein synthesis

## Abstract

Our previous studies found that bta-miR-106b and its corresponding target gene, *CDKN1A*, were differentially expressed between the mammary epithelium of lactating Holstein cows with extremely high and low milk protein and fat percentage, implying the potential role of bta-miR-106b in milk composition synthesis. In this study, with luciferase assay experiment, bta-miR-106b was validated to target the 3′-untranslated region (UTR) of bovine *CDKN1A*, thereby regulating its expression. Moreover, in bovine mammary epithelial cells (BMECs), over-expression of bta-miR-106b significantly down-regulated the *CDKN1A* expression at both mRNA and protein levels, and inhibitors of bta-miR-106b increased *CDKN1A* expression. Of note, we observed that bta-miR-106b accelerated cell proliferation and cell cycle, and changed the expressions of protein synthesis related pathways such as JAK-STAT and PI3K/AKT/mTOR through regulating *CDKN1A* expression. Our findings highlight the important regulatory role of bta-miR-106b in milk protein synthesis by targeting *CDKN1A* in dairy cattle.

## 1. Introduction

Bovine milk with optimal ingredients for newborn healthy growth and development is a high nutritional value food that is rich in protein and fat [1,2]. The milk composition, as an important indicator of milk quality, is one of the breeding objectives in dairy cattle. Previous studies have revealed many candidate genes involved in milk protein and fat synthesis and metabolism [3,4,5,6,7,8,9,10]. MicroRNAs, as a class of small RNAs, are involved in a great variety of biological pathways, such as development, differentiation, and metabolism, by regulating the expression of specific target gene mRNAs [11,12]. For the mammary tissue, some microRNAs related to mammary gland development, cell proliferation, and milk composition synthesis have been identified [13,14,15,16]. Bta-miR-15a inhibited the expression of casein and reduced the viability of mammary epithelial cells by targeting the *GHR* gene [17]. Bta-miR-181a regulated milk fat biosynthesis by targeting the *ACSL1* gene [18]. Bta-miR-208b can regulated cell cycle and promotes skeletal muscle cell proliferation by targeting the *CDKN1A* gene [19]. Mammary gland epithelial cell proliferation was inhibited with MiR-221 by targeting *STAT5a* and *IRS1* in PI3K/AKT/mTOR and JAK-STAT signaling pathways [20]. In BMECs, a study reported that the PI3K/AKT/mTOR and JAK-STAT signaling pathways regulated milk protein synthesis [21,22,23]. The PI3K/Akt/mTOR pathway was one of the most important regulatory pathways for casein synthesis [24,25]. The *mTOR* could be activated by the status of extracellular hormones and amino acids, which then activated the downstream molecules (*S6* and *4EBP1)*, and these could participate in the protein synthesis [26,27,28].

In our previous RNA sequencing study, bta-miR-106b and one of its target genes, *CDKN1A*, have been found to be differentially expressed between the mammary glands from lactating Holstein cows with extremely high and low milk protein and fat percentages [29,30]. We also found that the *CDKN1A* gene displayed significant genetic effects on milk protein yield and percentage in Chinese Holstein [31]. It has been reported that *CDKN1A* was a critical regulator of cell cycle [32,33] and bta-miR-106b plays an important role in cell cycle by directly targeting *CDKN1A* [34,35]. Bta-miR-106b belongs to the bta-miR-106b-25 cluster (containing bta-miR-106b, miR-93, and miR-25) [36] and plays important roles in cancer [37], cell differentiation [38], cell proliferation processes [39], cell cycle [34], and breast cancer [40]. Therefore, the purpose of the present study was to validate the regulatory roles of bta-miR-106b in cell proliferation, cell cycle, and protein and fat synthesis by targeting the *CDKN1A* gene in BMECs.

## 2. Materials and Methods

### 2.1. Target Gene Prediction and Luciferase Assays

The mature sequence of bta-miR-106b was obtained from the miRBase Sequence Database (http://www.mirbase.org, accessed on 25 September 2019), and target genes were predicted by scanning the seed regions between bta-miR-106b and the corresponding genes with TargetScan (http://www.targetscan.org, accessed on 25 September 2019).

To identify the interactions between bta-miR-106b and target genes, 3′UTR fragments of the targets were amplified by PCR and ligated into the pmirGLO Dual-Luciferase microRNAs Target Expression Vector (Promega) using NheI and XhoI restriction enzymes. The full-length *CDKN1A* 3′UTR sequence was transfected into HEK293 cells (Procell, Hubei, China). Cells were cultured at 37 °C with 5% CO_2_ in DMEM supplemented with 5% fetal bovine serum (Invitrogen, Carlsbad, CA, USA). Before transfection, HEK293 cells were plated into 24-well plates at 1.0 × 10^5^ cells/well for 24 h. Then, 30 ng pmirGLO-CDKN1A-3′UTR with 50 μL opti-MEM (Invitrogen) and 30 nM (final concentration) mimics, inhibitor, and control of bta-miR-106b were co-transfected into each well with 1 μL Lipofectamine 2000 (Invitrogen), respectively. Relative firefly luciferase activities (normalized to Renilla luciferase activities) were measured 24 h after transfection with the Dual-Luciferase Reporter Assay Kit (Promega) on TECAN Infinite 200 multifunctional microplate reader. All experiments were performed in triplicate with data averaged from at least three independent experiments.

Partial sequence of the *CDKN1A* 3′UTR was as follows; the underscore is a seed sequence bound to bta-miR-106b:

TACGAAGTCCTGCCCCCTCTTTCCTGCATTCTCAGACCTGAATTCCTTACAATCTGAGAAGTAAATAGATAGCACTTTGAAGGGGGCCCAACAGAGTTGGAGGCGTCATCAAAACTTTGGGTTCCCCTTGCCCCCTCTAAGGTTGGGCAGGGTGACCCTGAAGTGGGCACAGCCTGGAAATGGGGCTGGGGGATTGGACACCCT

### 2.2. BMECs Isolation and Culture

Healthy lactating Chinese Holstein cows from the Beijing Sanyuan Dairy Farming Center were harvested in the same harvest facility. The BMECs were isolated, cultured, and identified as previously described [41]. After cows were harvested, mammary parenchyma tissues were aseptically removed from the mammary glands and mixed, and were washed with sterilized saline several times and cut into small pieces of about 1–2 mm^3^. These small pieces were cultivated in DMEM/F12, with the rat tail collagen overlaid on the bottom layer of the tissue culture dishes. All of the cells were maintained at 37 °C in an atmosphere containing 5% CO_2_. According to the difference in trypsin tolerance between BMECs and fibroblasts, the fibroblasts were digested with 0.05% trypsin, and then BMECs were digested with 0.25% trypsin. The pure mammary epithelial cells were isolated after passages 3 to 4. Cytoskeletal structure of the cultured BMECs was evaluated with cytokeratin 18 immunofluorescence and DAPI dying. Only passages 3 to 4 of BMECs were used in this study.

### 2.3. Transfections of Bta-miR-106b to BMECs

Mimics and inhibitor for bta-miR-106b were designed and synthesized by GenePharma (ShangHai, China). The BMECs were seeded at 3 × 10^5^ cells per well in a six-well plate and grew for 24 h. Then, the cells were transfected with 50 pmol of bta-miR-106b mimics or negative control (mimics-NC) and 100 pmol of bta-miR-106b inhibitor or negative control (inhibitor-NC) using Lipofectamine 2000 (Invitrogen), respectively. Total RNA and protein were isolated from the transfected cells 48/72 h post-transfection for further analysis.

### 2.4. Cell Proliferation Assay and Cell Cycle Analysis of BMECs

Cell proliferation was assessed using a CCK-8 Cell Counting kit (Beyotime, Shanghai, China) in a 96-well plate (2 × 10^3^ cells per well). A total of 10 µL of CCK-8 solution was added to 100 µL cell suspension and incubated for 2 h, followed by absorbance assessment at 450 nm (*n* = 5). All experiments were performed in triplicates for each transfection.

The Cell-Light EdU DNA cell proliferation kit (Beyotime, Beijing, China) was used to determine the proliferation rate of the cells. Briefly, the cells were incubated with 50 μM EdU for 2 h before fixation, permeabilization, and EdU staining. The cell nuclei were stained with DAPI at a concentration of 5 μg/mL for 30 min, and the cells were examined using a florescence microscope (Olympus, Tokyo, Japan).

Cell cycle was determined by flow cytometry, as described previously [42]. Treated cells were harvested and washed two times with cold PBS and then fixed with 1 mL of 70% ethanol overnight at 4 °C. Fixed cells were centrifuged for 3 min at 1200 g and then washed with cold PBS and resuspended in PBS with 50 mg/mL prepodium iodide (PI) and 1 mg/mL RNase. The stained cells were analyzed for DNA content by fluorescence-activated cell sorting (FACs) in a FACs Calibur (Becton Dickinson Instrument, San Jose, CA, USA). Cell cycle fractions were quantified using the CytExpert 2.0 software (Becton Dickinson).

### 2.5. Over-Expression and RNA Interference (RNAi) Analysis of CDKN1A Gene in BMECs

Based on the sequence of the bovine *CDKN1A* gene (GenBank accession number: NM_001098958.2), eukaryotic expression plasmid pEGFP-N1-CDKN1A was constructed using the pEGFP-N1-Vector (Clontech, BD Biosciences) with the mammary gland tissue cDNA as the template. BMECs were transfected with 1.0 μg/well of pEGFP-N1-CDKN1A (pCDKN1A) as well as the negative control of pEGFP-N1-Vector (vector) using the Lipofectamine 2000 transfection reagent. The total RNA and protein were isolated from the transfected cells 48 h post-transfection for further analysis. In the meantime, three interference fragments (siRNA-112, siRNA-563, and siRNA-669; Table 1) targeting the second and third exons of *CDKN1A* were designed with NCBI and synthesized by GenePharma (Suzhou, China). The siRNA fragments as well as negative control were transfected into BMECs, respectively.

### 2.6. Quantitative RT-PCR

Total RNA was extracted from cultured cells using Trizol reagent (TaKaRa, Dalian, China). Reverse transcription was carried out using PrimeScript RT reagent Kit (TaKaRa). The mRNA expressions of *CDKN1A* and 24 genes related to protein and lipid synthesis were assessed by qPCR with LightCycler^®^ 480 SYBR Green I Master on a Roche 480 instrument (Roche Applied Science, Penzberg, Germany). Primer sequences are described in Table 2. The expression levels of bta-miR-106b and genes were normalized by *U6* and *GAPDH*, respectively. The results are representative of at least three independent experiments to determine the statistical significance.

### 2.7. Western Blot Analysis

Total protein was extracted from the BMECs using RIPA (radio immune-precipitation assay) cell lysis solution (Beyotime, Nanjing, China) after transfection for 72 h. The samples were separated on a 10% SDS-PAGE gel (30 μg of protein per sample) and transferred onto nitrocellulose membranes (Bio-Rad, Shanghai, China). After blocking with 5% skim milk powder solution for 1 h at 37 °C, the membranes were incubated overnight at 4 °C with a murine polyclonal rabbit antibody against CDKN1A, STAT5, p-STAT5, PI3K, AKT1, p-AKT1, mTOR, p-mTOR, and CSN2 (Santa Cruz, CA, USA), and then incubated with the horseradish peroxidase (HRP)-conjugated secondary antibody (Beyotime, Beijing, China). The protein binds were visualized with Super ECL Plus (Solarbio, Beijing, China), and GAPDH (Abcam) was used as an internal reference control. The results reported represent the mean of three independent experiments.

### 2.8. Statistical Analysis

Data were presented as the means ± standard deviation of at least three independent experiments or animals. Statistical differences among groups were compared with one-way analysis of variance and the differences between pair-designed experiments were compared with Student’s *t*-tests using GraphPad Prism 7.0 software (San Diego, CA, USA). Statistical significance was declared as *p* < 0.05 (*), *p* < 0.01 (**), and *p* < 0.001 (***).

## 3. Results

### 3.1. Bta-miR-106b Targeted the 3′UTR of the CDKN1A Gene

Based on the miRBase database, the CDKN1A gene that contains the specific binding sites for bta-miR-106b was predicted as the target using TargetScan 6.2 software (Figure 1A). Further, the pmirGLO vectors, carrring the 3′UTR sequence of CDKN1A, were transferred into HEK293T cells with bta-miR-106b mimics and inhibitor, respectively. It was observed that cotransfection of bta-miR-106b mimics and CDKN1A-3′UTR significantly decreased the luciferase signal at 48 h after transfection (Figure 1B; *p* < 0.01), while no significant change was observed in other transfection groups. These results demonstrated the regulatory role of bta-miR-106b on CDKN1A expresion by targeting its 3′UTR.

### 3.2. Regulation of Bta-miR-106b on CDKN1A Expression in BMECs

The primary BMECs were isolated from acinous fragments of fresh mammary gland epithelium of lactating Holstein cows. On the 8th day of culture, the BMECs appeared as polygonal shaped and were mixed with a few fibroblasts. After culturing for 15 days, the cells grew densely and then were purified with trypsin digestion to eliminate fibroblasts. By performing keratin 18 (green) staining, the BMECs presented integrated cytoskeletal structures (Figure 2), indicating their morphological and functional normality during cultures.

The BMECs were transfected with bta-miR-106b mimics and inhibitors and incubated for 48 h, and the same was carried out with negative controls, respectively. With qPCR, it was observed that supplementation of 50 pmol/L mimics significantly increased the mRNA level of bta-miR-106b (Figure 3A) and supplementation of 100 pmol/L inhibitor resulted in a significant decrease in bta-miR-106b expression compared with negative controls in BMECs (Figure 3B).

Of note, the mRNA expression of the CDKN1A gene was significantly decreased after transfection with bta-miR-106b mimics (Figure 3C; *p* < 0.01); at the same time, it was significantly increased in the bta-miR-106b inhibtor group (Figure 3D; *p* < 0.05), confirming the regulation of bta-miR-106b on CDKN1A expresion. With Western blot analysis, the same expression trends of CDKN1A gene were found at the protein level (Figure 3E,F).

### 3.3. Bta-miR-106b Promoted the Proliferation and Cell Cycle of BMECs

With the CCK-8 assay, it was shown that the number of viable mammary epithelial cells in the bta-miR-106b mimics treatment group was significantly greater than that in negative controls after transfection of 48 h (Figure 3G); in contrast, fewer live cells were observed in the inhibitor group (Figure 3H). Then, a rescue experiment was performed by over-expression of the *CDKN1A* gene in BMECs with pEGFP-N1- CDKN1A. With EDU assay, elevation of *CDKN1A* expression significantly decreased the proliferation ability of BMECs compared with the bta-miR-106b mimics group (Figure 3I), confirming *CDKN1A* was a down-stream target for bta-miR-106b.

Additionally, it was observed that transfection bta-miR-106b mimics significantly decreased the BMECs in G0/G1 phase and increased the cells in S phase with flow cytometry analysis (Figure 4A; *p* < 0.01); conversely, bta-miR-106b inhibitors led to a significant increase in cells in the G0/G1 phase and a decrease in cells in the S phase (Figure 4B; *p* < 0.05). Further, the expression levels of cell-cycle-related genes such as *CDK4*, *Cyclin E*, and *CDK2* were observed to be up-regulated when bta-miR-106b was over-expressed (Figure 4C; *p* < 0.05); on the contrary, bta-miR-106b inhibitors decreased the expression of *Cyclin D* and *CDK2* genes (Figure 4D; *p* < 0.05, *p* < 0.01).

### 3.4. Bta-miR-106b Regulated Milk Protein Synthesis through the PI3K/AKT/mTOR Pathways

To further investigate whether or not the bta-miR-106b regulates milk protein and fat synthesis, the expression of 16 functional genes related to lipid synthesis and 8 genes for protein synthesis in BMECs was evaluated by qPCR. As a result, over-expression of bta-miR-106b merely down-regulated the GAPM expression (Figure 5A; *p* < 0.05) and bta-miR-106b inhibitor merely up-regulated the SCD expression (Figure 5B; *p* < 0.01), suggesting that bta-miR-106b had little effect on milk fat synthesis and metabolism.

On the other hand, upon over-expression of bta-miR-106b, the mRNA levels of five important genes involved in JAK-STAT and PI3K/AKT/mTOR pathways (*JAK*, *STAT5*, *AKT1*, *S6*, and *4EBP1*) and the most important milk gene, *CSN2*, were significantly increased (Figure 5C; *p* < 0.05, *p* < 0.01). As expected, bta-miR-106b inhibitors significantly decreased the expression levels of *JAK2*, *PI3K*, *mTOR*, and *S6* (Figure 5D; *p* < 0.05, *p* < 0.01). Consistently, total and phosphorylated protein levels of *AKT1*, *mTOR*, and *PI3K* were markedly increased in the mimics group (Figure 5E,F) and decreased in the inhibitor group as well (Figure 5E,G). These results indicated the regulatory role of bta-miR-106b in milk protein synthesis.

### 3.5. CDKN1A Regulated Cell Cycle and Milk Protein Synthesis through PI3K/AKT/mTOR Pathways of BMECs

Furthermore, RNA interferen and over-expression experiments were performed to validate whether the CDKN1A regulates milk protein and fat synthesis. First, 48 h after transfection of siRNA-669 to BMECs with interference efficiency of greater than 70% (Figure 6A), knocked-down expression of CDKN1A decreased the BMECs in the G0/G1 phase and increased the cells in the S phase (Figure 6B). Simultaneously, Cyclin D, CDK4, Cyclin E, and CDK2 were significantly promoted (Figure 6C; *p* < 0.05, *p* < 0.01). In addition, down-regulation of CDKN1A significantly increased the expressions of STAT5, PI3K, AKT1, and mTOR genes in the JAK-STAT and PI3K/AKT/mTOR pathways, as well as the CSN2 gene (Figure 6D–F; *p* < 0.05, *p* < 0.01).

Conversely, over-expression of CDKN1A led to an increase in cells in the G0/G1 phase and a decrease in cells in the S phase (Figure 6G). A decrease in the expressions of Cyclin D, CDK4, CDK6, Cyclin E, and CDK2 genes was observed as well (Figure 6H; *p* < 0.05, *p* < 0.01). As expected, up-regulation of CDKN1A significantly decreased the expression levels of JAK2, PI3K, AKT1, mTOR, and S6 (Figure 6E,I,J; *p* < 0.05, *p* < 0.01).

## 4. Discussion

We previously found that bta-miR-106b and its target, the *CDKN1A* gene, showed significantly differential expression between the lactating mammary glands of Holstein cows with extremely high and low milk composition [29,30]. In this study, we further validated the regulatory role of bta-miR-106b on *CDKN1A* expression, thereby promoting mammary gland epithelial cell proliferation and activating the PI3K/AKT/mTOR pathways to regulate milk protein synthesis.

The development of the mammary gland is a complex process involving a large number of biological responses and signal transduction pathways. Important milk components such as protein, lactose, and fat were synthesized in the mammary gland [43,44,45]. MicroRNAs, small RNA, performed multiple functions by regulating the expression of complementary mRNA [46]. It has been reported that miR-106b promoted cell proliferation and cell cycle progression by targeting the *CDKN1A* gene in human and mouse [34,35,38,39,47,48,49]. In this study, we observed that the DNA synthesis level was significantly increased in BMECs after being transfected with bta-miR-106b mimics; especially, the over-expression of bta-miR-106b increased the cell proportion in the S phase while decreasing the cell proportion in the G1/G0 phase, presenting the regulatory role of bta-miR-106b in BMEC proliferation and cell cycle. In addition, our luciferase assay and rescue experiment also demonstrated the regulatory role of bta-miR-106b on *CDKN1A* expresion by targeting its 3′UTR. These results were consistent with the previous studies.

*CDKN1A* is a p53-inducible protein that was a critical regulator of cell survival and cell cycle by inhibiting the DNA synthesis regulator proliferation, cell nuclear antigen, and activation of *cyclinD1-CDK4/6* complexes [32,33]. Hawke et al. reported that skeletal muscle regeneration of *CDKN1A*-/- mouse was significantly enhanced compared with wild type muscle [50]. Moreover, many studies have shown that the *CDKN1A* gene could activate PI3K/AKT signaling pathways. Studies reported that knockdown of *CDKN1A* increased cell proliferation through activating the PI3K/AKT pathway and miR-93 promoted cell proliferation, migration, and invasion by activating the PI3K/AKT pathway via inhibition of *CDKN1A* in human and mouse [51,52]. *CDKN1A* might be a relevant *PDK2* responsible for AKT phosphorylation [53]. PI3K/AKT/mTOR and the JAK/STAT signaling pathways have been shown to regulate cell proliferation and milk protein synthesis in BMECs and mouse [21,22,23,24,25,54,55].

Similarly, in this study, we observed that knock-down of *CDKN1A* gene regulates cell cycle and increases genes’ expressions in PI3K/AKT/mTOR pathways (*AKT1, S6, 4EBP1*, and *PI3K*) and *CSN2* in BMECs, while *CDKN1A* over-expression displayed opposite trends. Meanwhile, over-expression and inhibitors of bta-miR-106b experiments showed results consistent with those of the *CDKN1A* gene. Especially, our previous study found that the *CDKN1A* gene was significantly associated with milk protein yield and percentage traits in Chinese Holstein [31]. In summary, these findings led us to believe that bta-miR-106b could regulate *CDKN1A* expression, thereby modulating cell proliferation, cell cycle and milk protein synthesis of mammary epithelial cells by activating the PI3K/AKT/mTOR signaling pathway in dairy cattle. At the same time, we observed that bta-miR-106b showed less impact on milk fat synthesis. However, these findings need in-depth validation through in vivo experimentations.

## 5. Conclusions

Herein, we experimentally confirmed that bta-miR-106b could promote mammary gland epithelial cell proliferation by targeting *CDKN1A*, thereby activating the PI3K/AKT/mTOR pathway to regulate milk protein synthesis. As far as we are aware, this is the first study to illustrate the regulatory mechanism of bta-miR-106b in BMECs of dairy cattle.

## Figures and Tables

**Figure 1 genes-13-02308-f001:**
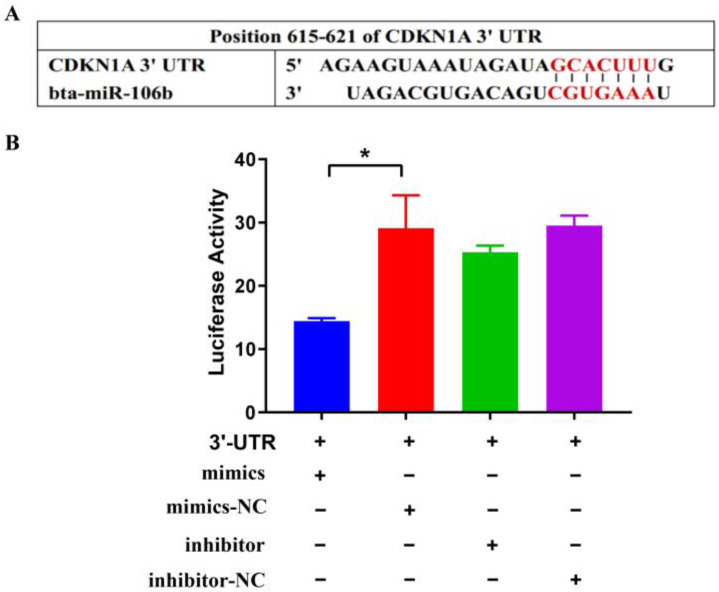
Bta-miR-106b regulated transcript activity by targeting CDKN1A 3′UTR with luciferase assays. (**A**) It was predicted Bta-miR-106b targeted the 3′UTR of bovine CDKN1A with TargetScan. (**B**) Luciferase activity was down-regulated after co-transfecting 3′UTR of CDKN1A with Bta-miR-106b mimics. (Note: * means *p* value < 0.05.)

**Figure 2 genes-13-02308-f002:**
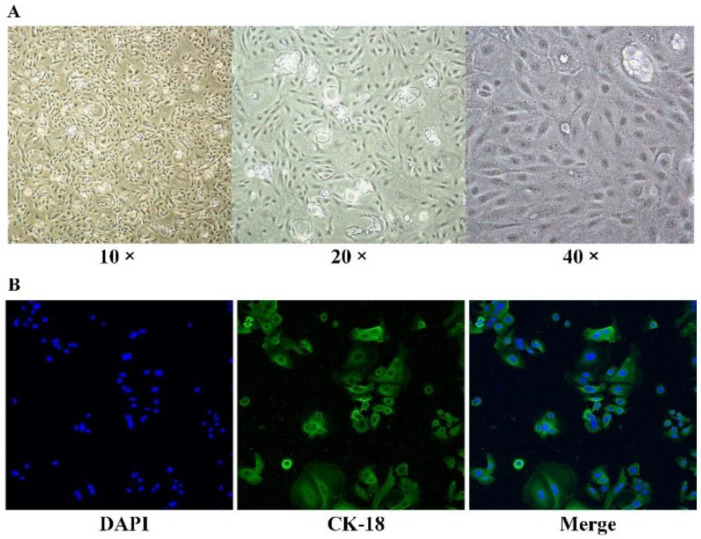
BMECs isolation. (**A**) Separation and purification of the BMECs. (**B**) CK-18 immunofluorescence and DAPI dying results of the BMECs (200×).

**Figure 3 genes-13-02308-f003:**
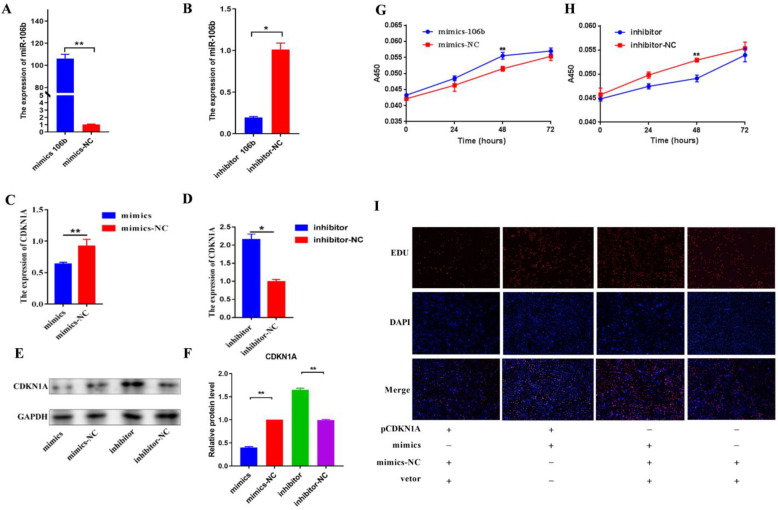
Bta-miR-106b regulated cell proliferation by targeting *CDKN1A* in BMECs. (**A**,**B**) The expression levels of bta-miR-106b at 48 h after transfection with bta-miR-106b mimics and inhibitors. (**C**,**D**) The mRNA levels of *CDKN1A* after transfection with bta-miR-106b mimics and inhibitors with qRT-PCR. (**E**,**F**) The protein expression of *CDKN1A* after transfection with bta-miR-106b mimics and inhibitors with Western blot. (**G**,**H**) The cell proliferation of BMECs at 0, 24, 48, and 72 h after transfection of bta-miR-106b mimics and inhibitors. (**I**) The proliferating BMECs were labeled with EdU. The click-it reaction revealed the EdU staining (red). The cell nuclei were stained with DAPI (blue). (Note: * means *p* value < 0.05; ** means *p* value < 0.01.).

**Figure 4 genes-13-02308-f004:**
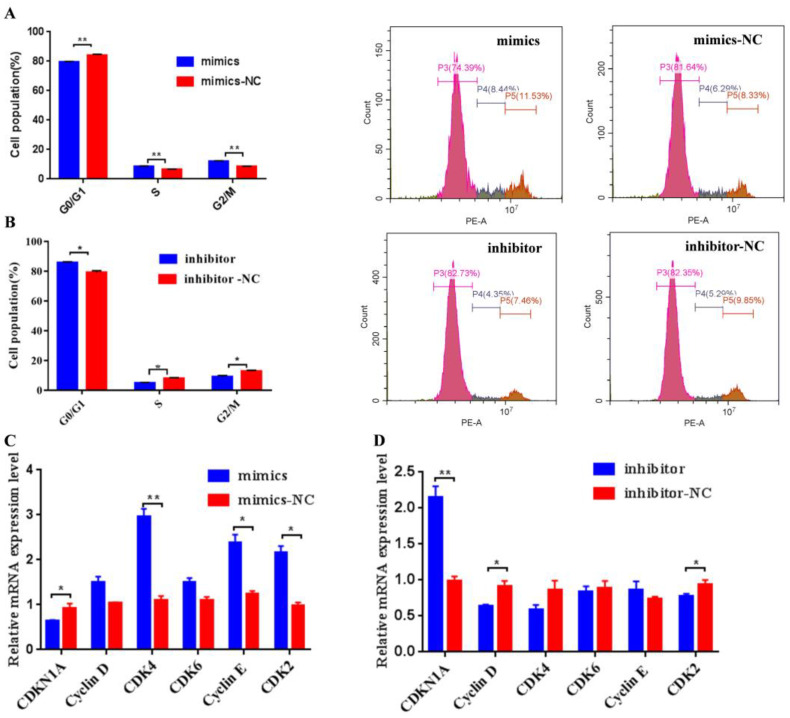
Bta-miR-106b promoted cell cycle progression from G0/G1 to the S phase in BMECs. (**A**) and (**B**) Distribution of cell cycle phases of BMECs with propidium iodide (PI) staining after transfection with bta-miR-106b mimics and bta-miR-106b inhibitors for 48 h. (**C**) and (**D**) The mRNA expression of cell cycle regulators specific to the G1/S phase in BMECs with bta-miR-106b mimics and inhibitors with qRT-PCR. (Note: * means *p* value < 0.05; ** means *p* value < 0.01.)

**Figure 5 genes-13-02308-f005:**
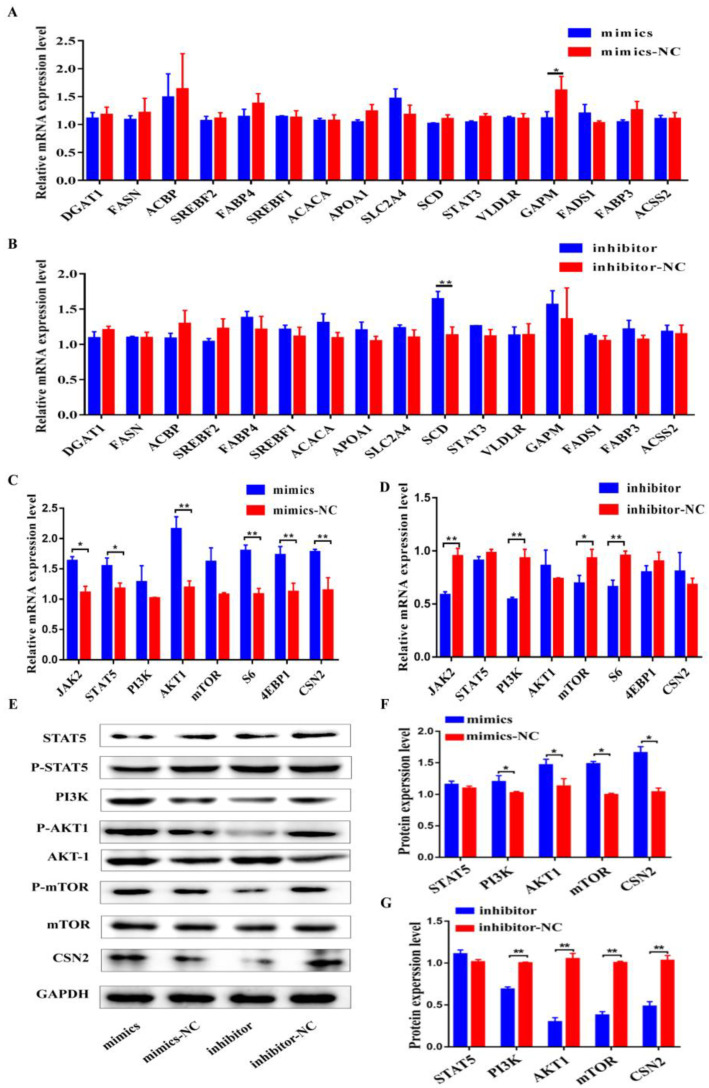
Regulation of Bta-miR-106b on the expressions of genes related to protein and fat synthesis in BMECs. (**A**,**B**) The mRNA levels of 16 genes involved in fat synthesis at 48 h after transfection with bta-miR-106b mimics and inhibitors in BMECs. (**C**,**D**) The mRNA levels of 8 genes involved in protein synthesis at 48 h after transfection with bta-miR-106b mimics and inhibitors in BMECs. (**E**–**G**) Western blot results and the protein expression levels of the genes involved in PI3K/AKT/mTOR pathway in BMECs at 48 h after transfection with bta-miR-106b mimics and inhibitors. (Note: * means *p* value < 0.05; ** means *p* value < 0.01.)

**Figure 6 genes-13-02308-f006:**
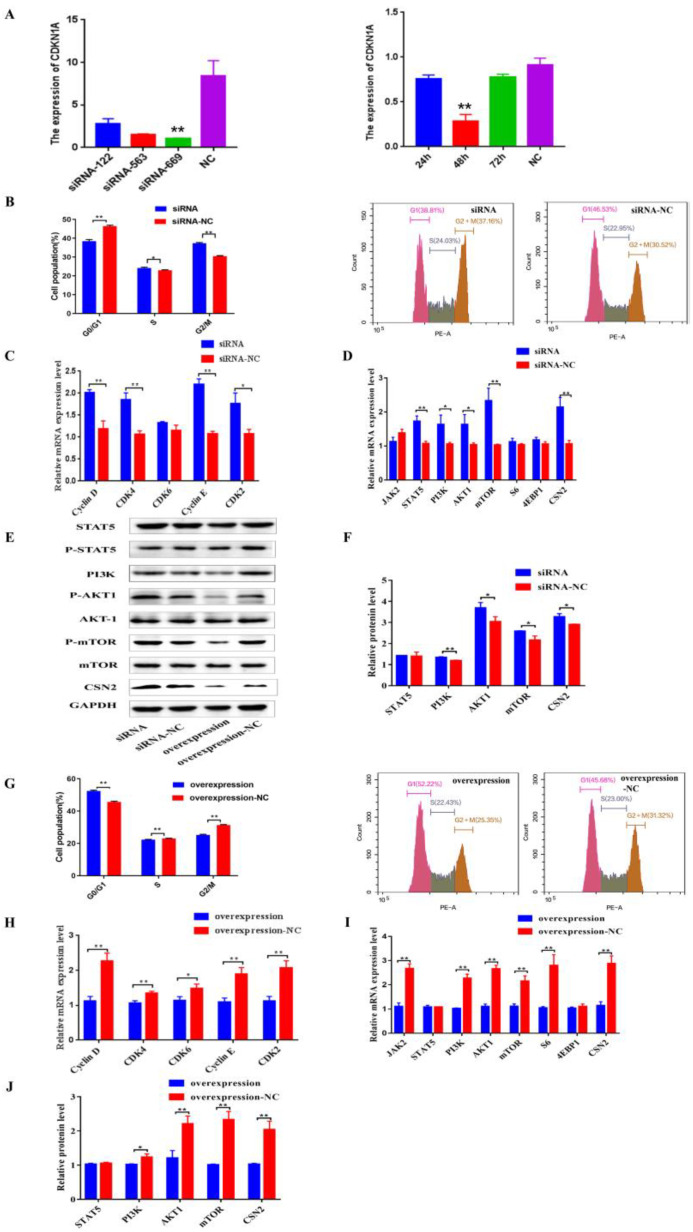
*CDKN1A* regulates BMECs cell cycle and milk protein synthesis through PI3K/AKT/mTOR pathways of BMECs. (**A**) Screening of the best inhibiting fragment and time of three siRNA. (**B**) Distribution of cell cycle phases of BMECs with PI staining after interference on *CDKN1A* for 48 h. (**C**) The mRNA expression of cell cycle regulators specific to G1/S phase in BMECs when *CDKN1A* was down-regulated with qRT-PCR. (**D**–**F**) The mRNA and protein expression levels of eight genes in protein synthesis after interference on *CDKN1A* in BMECs. (**G**) Distribution of cell cycle phases of BMECs with PI staining after over-expression of *CDKN1A*. (**H**) The mRNA expression of cell cycle regulators specific upon over-expression of *CDKN1A*. (**I**,**J**) The expression of eight genes involved in protein synthesis after up-regulation of *CDKN1A.* (Note: * means *p* value < 0.05; ** means *p* value < 0.01.)

**Table 1 genes-13-02308-t001:** Sequences for the siRNAs targeting the *CDKN1A* gene.

siRNAs	siRNA Sequences (5′-3′)	Target Gene Sequence (5′-3′)
siRNA-122	CCAGGGACGCGCAUCAAAUTT	CCAGGGACGCGCATCAAAT
	AUUUGAUGCGCGUCCCUGGTT	
siRNA-563	GCAGACUGAUCUGCUCCAATT	GCAGACTGATCTGCTCCAA
	UUGGAGCAGAUCAGUCUGCTT	
siRNA-669	CCUUCAGUUUGUGCGUCUUTT	CCTTCAGTTTGTGCGTCTT
	AAGACGCACAAACUGAAGGTT	

**Table 2 genes-13-02308-t002:** Primers for qRT-PCR.

Genes	Forward Primers	Reverse Primers
*CSN2*	AGTGAGGAACAGCAGCAAACAG	AGCAGAGGCAGAGGAAGGTG
*JAK2*	ACAGGGGCTGGCGTTCA	TATTGGTAACCAACAGCTCAAGG
*STAT5*	GTCCCTTCCCGTGGTTGT	CGGCCTTGAATTTCATGTTG
*mTOR*	ATGCTGTCCCTGGTCCTTATG	GGGTCAGAGAGTGGCCTTCAA
*S6K1*	CTGGGGAAGAGGTGCTTCAG	GTGCTCTGGTCGTTTGGAGA
*AKT1*	CCTGCCCTTCTACAACCAGG	GTCTTGGTCAGGTGGCGTAA
*PI3K*	AGCGCTGAGCAGTGTATCTT	GGGTATGGAGCCATCAGACG
*4EBP1*	CTGGGGACTACAGCACCAC	AGGTGATTCTGCCTGGCTTC
*CDKN1A*	GAGACCCCCAGAAGAGCCAC	AAAGTCGAAGTTCCACCGCT
*CDK2*	TTTGCTGAGATGGTGACCCG	TAACTCCTGGCCAAACCACC
*Cyclin D*	ATGAAGGAGACCATCCCCCT	CGCCAGGTTCCACTTGAGTT
*Cyclin E*	CGATGTCTCTGTTCGCTCCA	CCACACTGGCTTCTCACAGT
*CDK4*	GGCGAGGGTCTTCTCTGGT	CAGACGTCCATAAGCCTGACA
*CDK6*	GGCTCTTACCTCAGTGGTCG	TCGACATCTGAACTTCCACGA
*DGATI*	CCACTGGGACCTGAGGTGTC	GCATCACCACACACCAATTCA
*FASN*	ACCTCGTGAAGGCTGTGACTCA	TGAGTCGAGGCCAAGGTCTGAA
*ACBP*	AGGCTGATTTTGACAAGGCG	GATCTAACAGTGCTGGACACTCAATATC
*ACACA*	CATCTTGTCCGAAACGTCGAT	CCCTTCGAACATACACCTCCA
*SREBF1*	CCAGCTGACAGCTCCATTGA	TGCGCGCCACAAGGA
*SREBF2*	AGGTCTCTGGGCACCATGC	CATCACCGCAACCCCAAG
*FABP4*	TCCAGTGTGATGCGGTGTGTA	TGGATAGTGCAGCCAGTGTGA
*APOA1*	CGGCGGCTTCTCTTGTATAGC	TTCAAGCGTGAGCTGAAACG
*SLC2A4*	AAGCAAGTTGCCCATCCTCA	AAACTGTGGCTCCAATTTCGA
*STAT3*	TGACCGAGGTTGGAGGTTTG	TGGTCCACCTGATCATTCTGG
*FABP3*	GAACTCGACTCCCAGCTTGAA	AAGCCTACCACAATCATCGAAG
*ACSS2*	GGCGAATGCCTCTACTGCTT	GGCCAATCTTTTCTCTAATCTGCTT
*SCD*	TCCTGTTGTTGTGCTTCATCC	GGCATAACGGAATAAGGTGGC
*GAPM*	GCAGGTTTATCCAGTATGGCATT	GGACTGATATCTTCCTGATCATCTTG
*VLDLR*	GCCCAGAACAGTGCCATATGA	TTTTCACCATCACACCGCC
*FADS1*	GGTGGACTTGGCCTGGATG	TGACCATGAAGACAAGCCCC
*GAPDH*	GCTGCTTTTAATTCTGGC	CTTTCCATTGATGACGAG
*MARVELD1*	GCAGAAGUAUGGGGAAGCCTT	TCTGATCACAGACAGAGCACCAT
*ITGB4BP*	GAGGGCTGGTACATCCCAAG	CTCGCTGCCTCGGTTCAC

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
