# Peer review of "Bta-miR-106b Regulates Bovine Mammary Epithelial Cell Proliferation, Cell Cycle, and Milk Protein Synthesis by Targeting the CDKN1A Gene"

_genes, 2022, doi:10.3390/genes13122308_

Round 1

Reviewer 1 Report

This is an interesting and novel look at the role of CDKN1A in milk protein synthesis. Overall this appears to be a well designed and well written study. 

I have two minor concerns. First, I would change the language about animal harvest. I would replace the term dismembered with harvested, slaughterhouse with harvest facility, and slaughtered with harvested.  This term is less inflammatory and still describes the process.

Second, there is ample evidence that associations and mechanisms in cell culture systems do not always react similarly in in vivo systems. I would add a sentence acknowledging the limitation of this work until it is proven to function similarly in a whole animal system. 

Author Response

Dear Editor,

We appreciate the opportunity you provided us for a revision of our manuscript and the comments that the reviewers put forward. Following is our point-by-point reply to the reviewer comments, and the changes made were marked in the revised manuscript.

Q: First, I would change the language about animal harvest. I would replace the term dismembered with harvested, slaughterhouse with harvest facility, and slaughtered with harvested. This term is less inflammatory and still describes the process.

Response: Thank you so much for the positive comment and valuable suggestions. As suggested, we have replaced the term “dismembered” with “harvested”, “slaughterhouse” with “harvest facility”, and “slaughtered” with “harvested” (please see line 103).

Q: Second, there is ample evidence that associations and mechanisms in cell culture systems do not always react similarly in in vivo systems. I would add a sentence acknowledging the limitation of this work until it is proven to function similarly in a whole animal system.

Response: Thank you so much for the positive comment and valuable suggestions. As suggested, we have added “However, these findings need further in-depth validation through in vivo experiments.” at line 387.

Finally, we wish to thank you and your reviewers again for the valuable suggestions. If you have any question on this manuscript, please feel free to contact me. Thank you again for your time and favorable consideration.

Sincerely

Dongxiao Sun, Professor

Tel/Fax: +86-10-62734653 ; Email: [email protected]

Reviewer 2 Report

The study from Wu et al. validates their previous study from RNA-Seq using in vitro study. This study contributes merit to identifying the pathway of milk production. However, there are a few concerns that need to be clarified.

1#I checked your paper in the Turnitin program (file attached), I think you should paraphrase the previous publications, through the authors put the references. Please do not copy and paste, though they are your old works, please rewrite them.

2#In my opinion, the results in this paper can conclude that bta-miR-106b could promote mammary gland epithelial cell proliferation by targeting CDKN1A, but it is still far from “regulate milk protein synthesis” (Line 14, 317), though these two genes form their previous research on differentially expressed between the mammary epithelium of lactating Holstein cows with extremely high and low milk protein and fat percentage.

Introduction

I only found that the researcher mentioned a little about proliferation “MiR-221 inhibited mammary gland epithelial cell proliferation by targeting STAT5a and IRS1 in PI3K/AKT/mTOR and JAK-STAT signaling pathways”. They did not mention anything about why they needed to investigate the cell cycle. What is their hypothesis? Why do the authors target the proliferation pathway, but not the protein synthesis pathway directly? In addition, what is the objective of performing cell cycle analysis of BMECs? Please explain what your expectation is. And then wrote some points in the introduction, to give some background to the audience to understand your experimental design.

Materials and Methods

Line 87: Please provide the references for mammary epithelial cell isolation and the evidence that cytokeratin 18 is a robust marker to differentiate mammary epithelial cells and fibroblasts.

Line 101, 110,131, and 160: Since the passage number effect the proliferation, the authors should state the passage number clearly for each experiment. For example, which passage did the researchers perform “2.3 Transfections of bta-miR-106b to BMEC”, “2.5 Cell proliferation assay and cell cycle analysis of BMECs” and “Western blot analysis”?

Conclusion

Line 14, 317: Why do the authors conclude that bta-miR-106b and CDKN1A regulate milk protein synthesis, but fat percentage, since they find the differential expression for both traits from the previous study?

Author Response

Dear Editor,

We appreciate the opportunity you provided us for a revision of our manuscript and the comments that the reviewers put forward. Following is our point-by-point reply to the reviewer comments, and the changes made were marked in the revised manuscript.

Q: I checked your paper in the Turnitin program (file attached), I think you should paraphrase the previous publications, through the authors put the references. Please do not copy and paste, though they are your old works, please rewrite them.

Response: As suggested, we have re-written the paraphrase as following.

“MicroRNAs are a class of small RNAs that involved in a great variety of biological pathways, such as development, differentiation and metabolism, by regulating the expression of specific target gene mRNAs [11,12]. For the mammary tissue, some microRNAs related to mammary gland development, cell proliferation and milk composition synthesis have been identified [13-16]. Bta-miR-15a inhibited the expression of casein and reduced the viability of mammary epithelial cells by targeting the GHR gene [17]. Bta-miR-181a regulated milk fat biosynthesis by targeting the ACSL1 gene [18]. Bta-miR-208b regulated cell cycle and promotes skeletal muscle cell proliferation by targeting the CDKN1A gene [19]. Mammary gland epithelial cell proliferation was inhibited with MiR-221 by targeting STAT5a and IRS1 in PI3K/AKT/mTOR and JAK-STAT signaling pathways [20] (please see line 35, 42, 45)”.

“bta-miR-106b-25 cluster (containing bta-miR-106b, miR-93, and miR-25) [36] and plays important roles in cancer [37], cell differentiation [38], cell proliferation processes [39], cell cycle [34] and breast cancer [40]. Therefore, the purpose of the present study was to validate the regulatory roles of bta-miR-106b on cell proliferation, cell cycle, protein and fat synthesis by targeting the CDKN1A gene in BMECs (please see line 68, 70).”

“MicroRNAs, small RNA, performed multiple functions by regulating the expression of complementary mRNA [46].” “CDKN1A is a p53-inducible protein that was a critical regulator of cell survival and cell cycle.” “Studies reported that knockdown of CDKN1A increased cell proliferation through activating PI3K/AKT pathway and miR-93 promoted cell proliferation, migration and invasion by activating the PI3K/AKT pathway via inhibition of CDKN1A in human and mouse (please see line 349, 361, 367).”

Q: In my opinion, the results in this paper can conclude that bta-miR-106b could promote mammary gland epithelial cell proliferation by targeting CDKN1A, but it is still far from “regulate milk protein synthesis” (Line 14, 317), though these two genes form their previous research on differentially expressed between the mammary epithelium of lactating Holstein cows with extremely high and low milk protein and fat percentage.

Response: Thank you so much for the positive comment. Our previous studies (Cui et al, BMC Genomics 2014, 15, 226; Cui et al, Frontiers in genetics 2020, 11, 548268) found that bta-miR-106b and its corresponding target gene, CDKN1A, were differentially expressed between the mammary epithelium of lactating Holstein cows with extremely high and low milk protein and fat percentage, implying the potential role of bta-miR-106b on milk composition synthesis. In addition, some other studies have reported that JAK, STAT5, PI3K, AKT1, mTOR, S6, 4EBP1 genes are involved in milk protein synthesis (Yang et al, Journal of animal science 2008, 86, E36-50; Hayashi et al, Journal of dairy science 2009, 92, 1889-1899; Burgos et al, Journal of dairy science 2010, 93, 153-161; Toerien et al, J Nutr 2010, 140, 285-292; Appuhamy et al, Journal of dairy science 2014, 97, 419-429; Zhang et al, Molecules 2014, 19, 9435-9452; Gao et al, Journal of Zhejiang University. Science. B 2015, 16, 560-572; Luo et al, Scientific reports 2018, 8, 3912). These investigations suggested that bta-miR-106b may regulate milk compositions through targeting CDKN1A.

Therefore, the purpose of this study was to further investigate whether the bta-miR-106b regulates milk protein synthesis or not. Of note, we evaluated the expression levels of 8 genes (JAK, STAT5, PI3K, AKT1, mTOR, S6, 4EBP1 and CSN2) related to protein synthesis in BMECs, and found that the mRNA levels of 5 important genes involved in JAK-STAT and PI3K/AKT/mTOR pathways and CSN2 were significantly increased when bta-miR-106b was over-expressed (P < 0.05, P < 0.01), while, bta-miR-106b inhibitors significantly decreased the expression levels of these genes (P < 0.05, P < 0.01). Consistently, total and phosphorylated protein levels of AKT1, mTOR, and PI3K were markedly increased in mimics group and were decreased in inhibitor group as well. These results indicated the regulatory role of bta-miR-106b on milk protein synthesis (please see line 280).

Q: Introduction: I only found that the researcher mentioned a little about proliferation “MiR-221 inhibited mammary gland epithelial cell proliferation by targeting STAT5a and IRS1 in PI3K/AKT/mTOR and JAK-STAT signaling pathways”. They did not mention anything about why they needed to investigate the cell cycle. What is their hypothesis? Why do the authors target the proliferation pathway, but not the protein synthesis pathway directly? In addition, what is the objective of performing cell cycle analysis of BMECs? Please explain what your expectation is. And then wrote some points in the introduction, to give some background to the audience to understand your experimental design.

Response: Thank you so much for your comments and suggestions. In terms of the cell cycle, researches showed that CDKN1A, cyclin dependent kinase inhibitor 1A, was a critical regulator of cell cycle (Xiong et al, Nature 1993, 366(6456):701-704; Sherr and Roberts, Genes & development 1999, 13(12):1501-1512). Furthermore, several studies reported that bta-miR-106b plays an important role in cell cycle by directly targeting CDKN1A (Ivanovska et al, Molecular and cellular biology 2008, 28, 2167-2174; He et al, Experimental and therapeutic medicine 2020, 19, 3203-3210). Aa suggested, we have added “It has been reported that CDKN1A was a critical regulator of cell cycle [32,33], and bta-miR-106b plays an important role in cell cycle by directly targeting CDKN1A [34,35].” at line 66. Therefore, we wish to further investigate whether the bta-miR-106b regulates cell cycle and milk compositions through targeting CDKN1A. Aa suggested, we have added some statements about the regulation of cell cycle in the Introduction and Discussion section as following.

“Therefore, the purpose of the present study was to validate the regulatory roles of bta-miR-106b on cell proliferation, cell cycle, protein and fat synthesis by targeting the CDKN1A gene in BMECs.”; please see line 69).

“Bta-miR-208b regulated cell cycle and promotes skeletal muscle cell proliferation by targeting the CDKN1A gene [19].” “Bta-miR-106b plays important roles in cancer [37], cell differentiation [38], cell proliferation processes [39], cell cycle [34] (please see line 43, 68; reference at line 468, 510).”

On the other hand, it has been shown that PI3K/AKT/mTOR and JAK-STAT signaling pathways can not only regulate cell proliferation but also milk protein synthesis (Yang et al, Journal of animal science 2008, 86, E36-50; Hayashi et al, Journal of dairy science 2009, 92, 1889-1899; Burgos et al, Journal of dairy science 2010, 93, 153-161; Toerien et al, J Nutr 2010, 140, 285-292; Appuhamy et al, Journal of dairy science 2014, 97, 419-429; Zhang et al, Molecules 2014, 19, 9435-9452; Gao et al, Journal of Zhejiang University. Science. B 2015, 16, 560-572; Luo et al, Scientific reports 2018, 8, 3912; Jiao et al, Journal of dairy science 2019, 102, 426-435). Therefore, we have added some points about PI3K/AKT/mTOR and JAK-STAT pathway regulated milk protein synthesis in the Introduction section to the audience to understand our experimental design as following.

“In BMECs, study reported that PI3K/AKT/mTOR and JAK-STAT signaling pathway regulated milk protein synthesis [21-23]. The PI3K/Akt/mTOR pathway was one of the most important regulatory pathways for casein synthesis [24,25]. The mTOR could be activated by the status of extracellular hormones and amino acids and, then activated the downstream molecules (S6 and 4EBP1), and these could participate in the protein synthesis [26-28] (please see line 52-57; reference at line 474-495).”

Q: Materials and Methods: Line 87: Please provide the references for mammary epithelial cell isolation and the evidence that cytokeratin 18 is a robust marker to differentiate mammary epithelial cells and fibroblasts.

Response: Done as suggested, we have added a reference and the sentence was revised to “The BMECs were isolated, cultured and identified as previously described [41] (please see line 108; reference at line 528).”

Q: Materials and Methods: Line 101, 110,131, and 160: Since the passage number effect the proliferation, the authors should state the passage number clearly for each experiment. For example, which passage did the researchers perform “2.3 Transfections of bta-miR-106b to BMEC”, “2.5 Cell proliferation assay and cell cycle analysis of BMECs” and “Western blot analysis”?

Response: As suggested, we have added “Only passages 3 to 4 BMECs were used in this study.” at line 119.

Q: Conclusion: Line 14, 317: Why do the authors conclude that bta-miR-106b and CDKN1A regulate milk protein synthesis since they find the differential expression for both traits from the previous study?

Response: Thank you so much for your valuable comment. Our previous studies (Cui et al, BMC Genomics 2014, 15, 226; Cui et al, Frontiers in genetics 2020, 11, 548268) found that bta-miR-106b and its corresponding target gene, CDKN1A, were differentially expressed between the mammary epithelium of lactating Holstein cows with extremely high and low milk protein and fat percentage, implying the potential role of bta-miR-106b on milk composition synthesis.

In addition, some other studies have reported that JAK, STAT5, PI3K, AKT1, mTOR, S6, 4EBP1 and CSN2 genes are involved in milk protein synthesis (Yang et al, Journal of animal science 2008, 86, E36-50; Hayashi et al, Journal of dairy science 2009, 92, 1889-1899; Burgos et al, Journal of dairy science 2010, 93, 153-161; Toerien et al, J Nutr 2010, 140, 285-292; Appuhamy et al, Journal of dairy science 2014, 97, 419-429; Zhang et al, Molecules 2014, 19, 9435-9452; Gao et al, Journal of Zhejiang University. Science. B 2015, 16, 560-572; Luo et al, Scientific reports 2018, 8, 3912). These investigations suggested that bta-miR-106b may regulate milk compositions through targeting CDKN1A.

Therefore, the purpose of this study was to further investigate whether the bta-miR-106b regulates milk protein synthesis or not. Of note, we evaluated the expression of 8 genes (JAK, STAT5, PI3K, AKT1, mTOR, S6, 4EBP1 and CSN2) related to protein synthesis in BMECs, and found that the mRNA levels of 5 important genes involved in JAK-STAT and PI3K/AKT/mTOR pathways and CSN2 were significantly increased when bta-miR-106b was over-expressed (P < 0.05, P < 0.01), while, bta-miR-106b inhibitors significantly decreased the expression levels of JAK2, PI3K, mTOR and S6 (P < 0.05, P < 0.01). Consistently, total and phosphorylated protein levels of AKT1, mTOR, and PI3K were markedly increased in mimics group and were decreased in inhibitor group as well.

Based on these results, we concluded that bta-miR-106b plays the regulatory role on milk protein synthesis by targeting CDKN1A gene (please see line 280).

Finally, we wish to thank you and your reviewers again for the valuable suggestions. If you have any question on this manuscript, please feel free to contact me. Thank you again for your time and favorable consideration.

Sincerely

Dongxiao Sun, Professor

Tel/Fax: +86-10-62734653 ; Email: [email protected]
